Tissue-specific expression of senescence biomarkers in spontaneously hypertensive rats: evidence of premature aging in hypertension

Somsura Ratthapon 1 2
Kamkajon Kanokwan 3 4
Chaimongkolnukul Khuanjit 5
Chantip Surachai 5
Teerapornpuntakit Jarinthorn 4 6
Wongdee Kannikar 4 7
Kamonsutthipaijit Nuntaporn 8
http://orcid.org/0000-0001-5778-2566 Tangtrongsup Suwimol 3 4
http://orcid.org/0000-0001-5356-1942 Panupinthu Nattapon 3 4 nattapon.pan@mahidol.ac.th
Tiyasatkulkovit Wacharaporn 1 wacharaporn.ti@chula.ac.th
Charoenphandhu Narattaphol 3 4 9 10
1 Department of Biology, Faculty of Science, Chulalongkorn University , Bangkok , Thailand
2 Master of Science Program in Zoology, Department of Biology, Faculty of Science, Chulalongkorn University , Bangkok , Thailand
3 Department of Physiology, Faculty of Science, Mahidol University , Bangkok , Thailand
4 Center of Calcium and Bone Research (COCAB), Faculty of Science, Mahidol University , Bangkok , Thailand
5 National Laboratory Animal Center, Mahidol University , Nakhon Pathom , Thailand
6 Department of Physiology, Faculty of Medical Science, Naresuan University , Phitsanulok , Thailand
7 Faculty of Allied Health Sciences, Burapha University , Chonburi , Thailand
8 Synchrotron Light Research Institute (Public Organization) , Nakhon Ratchasima , Thailand
9 Institute of Molecular Biosciences, Mahidol University , Nakhon Pathom , Thailand
10 The Academy of Science, The Royal Society of Thailand , Dusit, Bangkok , Thailand
Gould Gwyn
Electronic publication date: 2023 Oct 19
Publication date: 2023
Volume: 11
Electronic Location ID: e16300
Received 2023 Jun 27; Accepted 2023 Sep 25
Copyright: © 2023 Somsura et al.
Copyright year: 2023
Copyright holder: Somsura et al.
License: This is an open access article distributed under the terms of the Creative Commons Attribution License, which permits unrestricted use, distribution, reproduction and adaptation in any medium and for any purpose provided that it is properly attributed. For attribution, the original author(s), title, publication source (PeerJ) and either DOI or URL of the article must be cited.
License URL: https://creativecommons.org/licenses/by/4.0/

Keywords: Hypertension, Cellular senescence, Cyclin-dependent kinase inhibitor (CDKI), Senescence-associated secretory phenotype (SASP), Spontaneously hypertensive rat (SHR)

Funding: Mahidol University–Postdoctoral Grant Distinguished Research Professor Grant, National Research Council of Thailand–Mahidol University Fundamental Fund: fiscal year 2022 and 2023 by National Science Research and Innovation Fund (NSRF) Prasert Prasarttong-Osoth Research Grant Graduate School Thesis Grant, Chulalongkorn University Postdoctoral fellowship grant by Mahidol University The present study was supported by Mahidol University–Postdoctoral Grant (to Narattaphol Charoenphandhu and Kanokwan Kamkajon), the Distinguished Research Professor Grant from National Research Council of Thailand–Mahidol University (to Narattaphol Charoenphandhu), Mahidol University (Fundamental Fund: fiscal year 2022 and 2023 by National Science Research and Innovation Fund (NSRF); to Nattapon Panupinthu and Narattaphol Charoenphandhu, respectively), the Faculty of Science, Mahidol University (CIF/CNI grant to Nattapon Panupinthu and Narattaphol Charoenphandhu), the Prasert Prasarttong-Osoth Research Grant (to Narattaphol Charoenphandhu), and Graduate School Thesis Grant, Chulalongkorn University (to Ratthapon Somsura). Kanokwan Kamkajon was a postdoctoral fellow awarded by Mahidol University. The funders had no role in study design, data collection and analysis, decision to publish, or preparation of the manuscript.

==============================
Background

Cellular senescence is an age-related physiological process that contributes to tissue dysfunction and accelerated onset of chronic metabolic diseases including hypertension. Indeed, elevation of blood pressure in hypertension coincides with premature vascular aging and dysfunction. In addition, onsets of metabolic disturbance and osteopenia in patients with hypertension have also been reported. It is possible that hypertension enhances premature aging and causes progressive loss of function in multiple organs. However, the landscape of cellular senescence in critical tissues affected by hypertension remains elusive.

Materials and Methods

Heart, liver, bone, hypothalamus, and kidney were collected from spontaneously hypertensive rats (SHR) and age- and sex-matched normotensive Wistar rats (WT) at 6, 12, 24 and 36 weeks of age (n = 10 animals/group). Changes in mRNA levels of senescence biomarkers namely cyclin-dependent kinase (CDK) inhibitors (CDKIs), i.e., Cdkn2a (encoding p16Ink4a) and Cdkn1a (encoding p21cip1) as well as senescence-associated secretory phenotypes (SASPs), i.e., Timp1, Mmp12, Il6 and Cxcl1, were determined. Additionally, bone collagen alignment and hydroxy apatite crystal dimensions were determined by synchrotron radiation small- and wide-angle X-ray scattering (SAXS/WAXS) techniques.

Results

Real-time PCR revealed that transcript levels of genes encoding CDKIs and SASPs in the heart and liver were upregulated in SHR from 6 to 36 weeks of age. Expression of Timp1 and Cxcl1 was increased in bone tissues isolated from 36-week-old SHR. In contrast, we found that expression levels of Timp1 and Il6 mRNA were decreased in hypothalamus and kidney of SHR in all age groups. Simultaneous SAXS/WAXS analysis also revealed misalignment of bone collagen fibers in SHR as compared to WT.

Conclusion

Premature aging was identified in an organ directly affected by high blood pressure (i.e., heart) and those with known functional defects in SHR (i.e., liver and bone). Cellular senescence was not evident in organs with autoregulation of blood pressure (i.e., brain and kidney). Our study suggested that cellular senescence is induced by persistently elevated blood pressure and in part, leading to organ dysfunction. Therefore, interventions that can both lower blood pressure and prevent cellular senescence should provide therapeutic benefits for treatment of cardiovascular and metabolic consequences.

Introduction

Premature aging is characterized by structural and functional changes in tissues that are caused by accumulations of oxidative stress, inflammatory mediators, and senescence cells (Lopez-Otin et al., 2013). Cellular senescence is used to describe cells that undergo irreversible growth arrest that can be triggered by several stressors associated with aging pathologies, including telomere erosion, replicative stress, DNA damage, reactive oxygen species (ROS), and oncogene activation (Fig. 1A). One of the important hallmarks of early senescence cells is cell cycle arrest via induction and activation of p53/p21cip1 and p16Ink4a/pRB tumor suppressor pathways (He & Sharpless, 2017). These cyclin-dependent kinase (CDK) inhibitors (CDKIs) act as negative regulators of cell cycle progression leading to cellular senescence (Beck, Horikawa & Harris, 2020; Chandler & Peters, 2013; Kumari & Jat, 2021). Although these senescent cells are unable to divide and increase in numbers, they are highly resistant to apoptosis and remain metabolically active (González-Gualda et al., 2021). Tolerance to any clearance mechanisms thus causes accumulation of these cells leading to chronic inflammation and tissue dysfunction. Senescent cells have been found to accumulate exponentially with increasing chronological age in multiple tissues, supporting the hypothesis that senescence is involved in aging and is one of its key hallmarks (Kumari & Jat, 2021).

Figure 1 The mechanism of senescent cell progression and experimental timeline.

Diagram explains the mechanism of senescent cell progression from initiation to full senescence (A), and timeline of the present experiment (B), please see text for detail. CDKIs, cyclin-dependent kinase inhibitors; ROS, reactive oxygen species; SA-β-gal, senescence-associated β-galactosidase; SASPs, senescence-associated secretory phenotypes; SHR, spontaneously hypertensive rat; WT, normotensive wild-type rats; SBP, systolic blood pressure; DBP, diastolic blood pressure; MAP, mean arterial blood pressure.

Hypertension is a major health problem worldwide and is highly prevalent in the aging populations (Mills, Stefanescu & He, 2020). However, elevated blood pressure is closely associated not only with increased chronological aging but also with biological aging, (Afsar & Afsar, 2023) Increased blood pressure without proper control imposes structural alterations in several organs leading to organ dysfunction. Chronic hypertension in particular, damages structures and functions of major vascular beds, such as those of the heart, brain and kidneys (Suvila & Niiranen, 2022). Indeed, the incidence and severity of cardiovascular diseases from both human and animal studies demonstrate sex differences. Specifically, males are at a higher risk of developing cardiovascular disease compared to age-matched premenopausal women. Moreover, male hypertensive rats displayed greater levels of fibrosis in the heart and kidney compared to female rats (Ansari, Walton & Denton, 2023). However, the understanding mechanisms by which hypertension induces end-organ damages are not completely understood. Interestingly, persistent elevation of blood pressure in essential hypertension is linked to the presence of premature vascular aging preceding the chronological ages (Harvey, Montezano & Touyz, 2015; Nilsson, 2020; Wang & Bennett, 2012). Therefore, it is possible that chronic elevation of systemic blood pressure induces widespread cellular senescence in those affected organs.

Senescence cells are highly active in the production and secretion of inflammatory cytokines, chemokines and proteases collectively known as senescence-associated secretory phenotypes (SASPs) (Birch & Gil, 2020). The effects of these SASPs were highlighted in the pathogenesis of inflammatory diseases and cancer (Byun et al., 2015). Of those, interleukin (IL)-6—one of the most prominent cytokines in SASP—has been shown to cause DNA damage and oncogenic stress in keratinocytes, melanocytes, monocytes, fibroblasts, and epithelial cells (Coppé et al., 2010). Our previous work in spontaneously hypertensive rats (SHR), an animal model for essential hypertension, displayed bone loss due to an increased bone resorption (Tiyasatkulkovit et al., 2019). Whether bone loss in SHR was induced by the production of SASP remained unknown. Furthermore, association of systemic hypertension and defects in liver metabolism has been reported (Nakagami, 2023); however, the connection remains poorly understood. In this study, we explored expression profiles of senescence-related CDKIs and a subset of SASPs in blood-pressure regulating organs, i.e., heart, kidney and hypothalamus as well as affected end organs, i.e., bone and liver. Tissues from age- sex-matched SHR and normotensive wild-type (WT) were collected and analyzed.

Materials and Methods

Animals

Four-week-old male spontaneously hypertensive rats (SHR/KyoMlac or SHR) (n = 40) and age/sex-matched normotensive wild-type rats (WMN/NrsMlac or WT) (n = 40) were acquired from the National Laboratory Animal Center of Thailand (NLAC), Mahidol University. Both groups of animals were housed in polycarbonate shoe box cages (two animals per cage) and the polycarbonate shelters were provided as enrichment devices. They were acclimatized in a controlled environment of 21 ± 1 °C, under 12-h light-dark cycle, 50–60% relative humidity. General health status was checked daily by veterinarians at the Central Animal Facility, Faculty of Science, Mahidol University (MUSC-CAF)—an AAALAC (Association for Assessment and Accreditation of Laboratory Animal Care)-accredited facility. They were fed standard laboratory food containing 1.0% calcium, 0.9% phosphorus and 4,000 IU/kg vitamin D (Perfect Companion Group Co., Ltd., Bangkok, Thailand) and reverse-osmosis (RO) water ad libitum. In an experiment that demonstrated the effect of antihypertensive agent on Timp1 expression, 24-week-old male SHR rats were randomly divided into two groups, i.e., (1) the vehicle group that was subcutaneously injected with distilled water, and (2) the treated group that was subcutaneously injected with 0.1 mg/kg/day propranolol everyday over a 12-week period (n = 8 per group). The experimental protocols have been approved by the Institutional Animal Care and Use Committee (IACUC), Faculty of Science, Mahidol University (No: MUSC61-050-452 and MUSC64-010-559). All studies related to animals were performed in accordance with relevant guidelines and regulations, including the ARRIVE guideline (https://arriveguidelines.org).

Blood pressure measurement and sample collection

Animals were housed at MUSC-CAF until reaching experimental endpoints at the age of 6, 12, 24, and 36 weeks, animals were randomly selected (Fig. 1B). Prior to euthanasia, assessment of systolic, diastolic and mean arterial blood pressures (SBP, DBP and MAP, respectively) was performed using non-invasive tail-cuff method (CODA; Kent Scientific Corp., Torrington, CT, USA) (n = 10/group). Briefly, animals were restrained in the restraining and acclimatized for at least 5 min, thereafter the blood pressure measurement was performed. At the end of the experiments, all animals were euthanized by intraperitoneal injection of 5 mg/kg xylazine (X-lazine; L.B.S. Laboratory Ltd., Part, Bangkok, Thailand) and 40 mg/kg tiletamine/zolazepam (Zoletil™ 100; Virbac Laboratories, Carros, France). After euthanasia, blood, heart, liver, kidneys, whole brain and long bones were collected. Serum samples were separated from whole blood for further analyses using enzyme-linked immunosorbent assay (ELISA) technique. Freshly isolated organs were kept frozen at –80 °C for RNA isolation and analyses of gene expression.

Tissue preparation and RNA isolation

Tissues from left ventricles and left lobe of liver were selected. Long bones were flushed with ice-cold phosphate-buffered saline to remove residual cells within bone marrow spaces. Bone specimens were chopped into small pieces. Whole brain tissues were placed with dorsal surfaces up on the cutting block. Razor blades were placed in fixed position to obtain coronal sections. In each section, hypothalamus was located and dissected as previously described (Heffner, Hartman & Seiden, 1980). Specimens from the kidneys were obtained after the surrounding connective tissues and renal capsules were carefully removed, they were then chopped into small pieces. Heart, liver, hypothalamus, and kidney were homogenized for RNA isolation following a protocol previously described (Tiyasatkulkovit et al., 2019). Bone tissues were homogenized using a Precellys Evolution Super Homogenizer with ceramic beads according to the manufacturer’s protocol (Bertin Instruments, Montigny-le-Bretonneux, France). Total RNAs were extracted using TRIZol reagent (Invitrogen, Carlsbad, CA, USA). Total RNA quality was assessed by measuring the absorbance at 260 and 280 nm using a NanoDrop-2000c spectrophotometer (Thermo Fisher Scientific, Waltham, MA, USA) and the 260/280 ratio ranged between 1.8 and 2.0.

Quantitative real-time reverse-transcription polymerase chain reaction (qRT-PCR)

Synthesis of cDNA was performed by reverse transcription of 1 μg total RNA using iScript cDNA synthesis kit (Bio-Rad, Hercules, CA, USA). Two housekeeping genes were included, i.e., 18S rRNA for heart, liver, and kidney samples and β-actin for long bones and hypothalamus samples. Consistency of reverse-transcription reactions was evaluated with variation coefficient less than (5%; n = 10 for each group). qRT-PCR and melting-curve analyses were performed using SsoFast Eva Green Supermix (Bio-Rad, Hercules, CA, USA) and QuantStudio 3 RT-PCR System (Applied Biosystems, Foster City, CA, USA). The PCR reactions were set for 40 cycles. Each cycle was fixed at 95 °C for 60 s, 54–58 °C for 30 s, and 72 °C for 30 s. Annealing temperatures for all primer pairs are listed in Table 1. These primers have been validated for specificity and efficiency using conventional RT-PCR. Relative gene expression was calculated based on 2–ΔΔCt method (Livak & Schmittgen, 2001).

Table 1 Rattus norvegicus oligonucleotide sequences used for quantitative RT-PCR.

Gene	Accession no.	Primer (forward/reverse)	Product size (bp)	Annealing temperature (°C)	
Cdkn2a (p16Ink4a)	NM_031550.2	5′- CGTGCGGTATTTGCGGTATC - 3 ′
5′- CTAGTCTCGCGTTGCCAGAA - 3 ′	184	57	
5′- CTAGTCTCGCGTTGCCAGAA - 3 ′	
Cdkn1a (p21cip21)	NM_080782.4	5′- TTGTCGCTGTCTTGCACTCT - 3 ′
5′- GGCACTTCAGGGCTTTCTC - 3 ′	165	58	
5′- GGCACTTCAGGGCTTTCTC - 3 ′	
Timp1	NM_053819.1	5′- GACCTATAGTGCTGGCTGTG - 3 ′
5′- GATCGCTCTGGTAGCCCTTC - 3 ′	134	58	
5′- GATCGCTCTGGTAGCCCTTC - 3 ′	
Il6	NM_012589.2	5′- GCAAGAGACTTCCAGCCAGT - 3 ′
5′- AGCCTCCGACTTGTGAAGTG - 3 ′	145	54	
5′- AGCCTCCGACTTGTGAAGTG - 3 ′	
Mmp-12	NM_053963.2	5′- ACGAGAGCGAATTTGCTGAA - 3 ′
5′- GCCCAGTTACTTCTAGCCCA - 3 ′	150	58	
5′- GCCCAGTTACTTCTAGCCCA - 3 ′	
Cxcl1	NM_030845.2	5′- GGCAGGGATTCACTTCAAGA - 3 ′
5′- GCCATCGGTGCAATCTATCT - 3 ′	205	58	
5′- GCCATCGGTGCAATCTATCT - 3 ′	
Housekeeping genes	
β-actin	NM_031144	5′- CAGAGCAAGAGAGGCATCCT - 3 ′
5′- GTCATCTTTTCACGGTTGGC - 3 ′	185	54	
5′- GTCATCTTTTCACGGTTGGC - 3 ′	
18s RNA	DQ066896	5′- GTAACCCGTTGAACCCCATT - 3 ′
5′- CCATCCAATCGGTAGTAGCG - 3 ′	151	57	
5′- CCATCCAATCGGTAGTAGCG - 3 ′	
Note:

Cdkn2a, cyclin-dependent kinase inhibitor 2a; Cdkn1a, cyclin-dependent kinase inhibitor 2a; Timp1, tissue inhibitor of metalloproteinase 1; Il6, interleukin 6; Mmp12, matrix metallopeptidase12; Cxcl1, the chemokine (C-X-C motif) ligand 1.

Enzyme-linked immunosorbent assay (ELISA)

Levels of TIMP metallopeptidase inhibitor 1 (Timp1) in the serum were determined by ELISA using Quantikine enzyme-linked immunosorbent assays (catalog no. RTM100; R&D systems, Minneapolis, MN, USA). We performed quality assessment of this technique. The intra- and inter-assay coefficients of variation were 3.1% and 6.7%, respectively. In addition, the detection limit was 0.0035 ng/mL. In brief, serum samples were diluted by 200-fold with the Calibrator Diluents (Calibrator Diluent RD 5–17). Rat Timp1 standards at various concentrations were used to obtain the standard curve with r2 at 0.7921. Monoclonal antibody specific for rat Timp1 was used to pre-coat a microplate and serum samples were added for 2 h. Next, polyclonal antibody specific for rat Timp1 was added for 2 h at room temperature. Substrate solution was then added to each well and incubated for 30 min. The color reagents were used to stop reaction. The optical density was determined at 450 nm by microplate reader (model M965; Metertech, Taipei City, Taiwan). Serum Timp1 concentrations were expressed in ng/mL.

Determination of plasma levels of O-methylated metabolites

Plasma levels of O-methylated metabolites of the catecholamines, i.e., metanephrine, normetanephrine and 3-methoxytyramine, were determined by liquid chromatography tandem mass spectrometry (LC-MS/MS) at Ramathibodi Hospital, Mahidol University, Bangkok, Thailand.

Immunofluorescent staining for p16Ink4a

Liver was fixed overnight in 4% paraformaldehyde in 0.1 M phosphate-buffered saline (PBS). After being embedded in paraffin, the specimens were cut cross-sectionally into 4-µm sections. All sections were incubated in sodium citrate solution (pH 6.0) at 95 °C for 40 min and then blocked with 5% fetal bovine serum in PBS containing 0.1% Tween-20 at room temperature for 2 h. Sections were then incubated with anti-CDKN2A/p16Ink4a antibody (catalog no. 54210; Abcam, Waltham, MA, USA) with 1:250 dilution at 4 °C for overnight in moist chamber. They were then washed and incubated with goat anti-mouse IgG (catalog no. ab150116; Abcam, Waltham, MA, USA) with 1:500 dilution at room temperature for 1 h. Vectashield Plus antifade mounting medium (catalog no. H-1900; Vector Laboratories, Newark, CA, USA) containing DAPI at 1:5,000 dilution was applied, and fluorescent signals were visualized under a fluorescent microscope (Ni-U model eclipse, Nikon, Japan).

Synchrotron small-and wide-angle X-ray scattering (SAXS/WAXS)

Synchrotron—a cyclic electron accelerator (1.2 GeV) operated by Synchrotron Light Research Institute (SLRI), Nakhon Ratchasima, Thailand—was utilized to generate high-quality intense X-ray beams. To determine the collagen alignment of the bone, SAXS and WAXS measurements were performed at BL1.3W: SAXS/WAXS, SLRI. Fresh frozen femoral cortical bone specimens from SHR and WT at 24 weeks of age were collected. Each bone sample was held to the aluminum frame with Kapton tape during simultaneous SAXS/WAXS measurements. An exposure time of 180 s and X-ray wavelength of 1.38 Å were used in both SAXS and WAXS techniques. The distances to detector for SAXS and WAXS were 4,969 and 149 mm, providing q range of 0.072–0.725 nm–1 and 2θCu range of 20–55°, respectively, and these protocols were calibrated using SEBS block co-polymer for SAXS and 4-bromobenzoic acid for WAXS. The data were background subtracted by using Kapton tape measurement. The SAXS and WAXS scattering patterns were recorded with a MarCCD165 and LX170-HS detectors, respectively. The raw data were processed by the SAXSIT program, which was developed by SLRI staff to obtain scattering curves. 1D-SAXS analysis was used to characterize a d-spacing value, calculated through Bragg’s Law using the following equation:

d=2nπ/q

where d is d-spacing, n is the number of periods and q is the SAXS scattering curves of samples that were averaged to each of their corresponding types, i.e., WT and SHR, before calculating d-spacing.

The periodic structure of the arrangement of the intrafibrillar collagen molecules was determined by 2D-SAXS analysis. The 1D-SAXS curve was characterized by a d-spacing value. The mineral crystals of carbonated apatite, which contributed to diffraction peaks, were evaluated from WAXS curves. The crystal lengths along the c-axis were indicated from the full-width half maximum (FWHM) of the (002) reflection at 2θCu about 29.7° and the crystal widths in the ab-plane were determined from the (310) reflection which is perpendicular to (002) at 2θCu about 39.5° (Lange et al., 2011). Pseudo Voigt function was used to fit in each peak (Fig fitting) using FityK software (Wojdyr, 2010). Subsequently, the FWHM of the fitted peak was used to calculate the crystal dimensions using Scherrer’s equation (Patterson, 1939):

D=Kλ/(βcosθ)

where D is the average crystal size corresponding to the reflection; K is the Scherrer constant depending on the crystal shape (typically about 0.9); λ is the X-ray wavelength; β is the FWHM of the peak in radians 2θ located at scattering angle θ.

The instrumental broadening contribution of the hydroxy apatite crystallite was corrected by the previously reported method using the Lab-6 standard (Lange et al., 2011).

Histology of the aorta

After euthanasia, the aorta was collected and washed with ice-cold sterile normal saline solution, fixed in 4% paraformaldehyde for 12 h at 4 °C. Tissues were dehydrated and cleared by graded ethanol and xylene, respectively, and embedded in paraffin. Paraffin blocks were cut into 5-µm thickness and stained with hematoxylin and eosin (H & E) for histological examination. Slides were visualized under a light microscope (Ni-U model eclipse; Nikon, Tokyo, Japan) with cellSens image acquisition software (Olympus, Tokyo, Japan).

Transmission electron microscopy (TEM)

The freshly collected left ventricular cardiac tissue was analyzed for ultrastructural morphology of vascular endothelium. Cardiac tissue was fixed with 2.5% glutaraldehyde in 0.1 M phosphate buffered saline (PBS) at 4 °C for 24 h. The sample was washed three times in pre-chilled PBS at 4 °C for 5 min each. The post-fixation of samples was performed under darkness using 1% osmium tetroxide (OsO4) in 0.1 M PBS for 2 h and then washed three times with 0.1 M PBS for 5 min each. Cardiac tissue was dehydrated with graded ethanol series (70%, 80%, 90%, 95% twice and 100% three times for 30 min each). Cardiac tissue was dehydrated using graded ethanol series ranging from 70% to absolute ethanol with each step lasting 30 min. Subsequently, the samples were permeated twice with pure propylene oxide (PO) and embedded in Araldite 502 resin. The resin-embedded tissue blocks were then sliced into 60–70 nm in thickness. These ultra-thin sections were mounted on copper grids and subjected to staining with uranyl acetate for 45 min followed by 0.1% lead citrate for 45 min. Finally, the prepared samples were observed under a TEM (model Tecnai 20G2; Philips, Cambridge, MA, USA) at the Microscopic Center, Faculty of Science, Burapha University, Chonburi, Thailand.

Statistical analysis

The results are expressed as means ± standard errors of mean (SEM). The two-group data were analyzed by unpaired Student’s t-test. Unless otherwise specified, statistical significance was considered when P < 0.05. All data were analyzed by GraphPad Prism 9 (GraphPad, San Diego, CA, USA).

Results

SHR exhibited an elevation of blood pressure without systemic change in O-metabolites of catecholamines and Timp1 levels in the blood

First, we evaluated the model of systemic hypertension used in this study. Longitudinal assessment of blood pressure parameters in SHR revealed persistent elevation of all three parameters of blood pressure, i.e., systolic, diastolic and mean arterial pressure. Quantitative analysis showed that the elevations reached statistical significance as early as 6 weeks and persisted up to 36 weeks of age (Figs. 2A–2C). Our findings on the degrees of change in these parameters in SHR compared to those normotensive WT controls that characterized the model were consistent with previous report (Okamoto & Aoki, 1963). In addition, SHR manifested thickening of the smooth muscle layer and enlargement of the aorta as compared to WT (Figs. S1A and S1B). Nevertheless, the endothelium and underlying intimal layers remained intact without apparent histological abnormality. TEM analysis of the cardiac vessels also showed that the endothelial cells of SHR were apparently normal (Fig. S1C).

Figure 2 Blood pressure profiles in SHR.

Blood pressure profiles, (A) systolic blood pressure (SBP), (B) diastolic blood pressure (DBP), and (C) mean arterial blood pressure (MAP) of spontaneously hypertensive (SHR) and wild-type (WT) rats at the age of 6, 12, 24 and 36 weeks. Results are presented as mean ± SEM. *P < 0.05; **P < 0.01; ***P < 0.001 compared with age-matched WT group.

Chronic elevation in blood pressure found in SHR led us to ask whether there was a progressive increase in sympathetic activation and SASP production in these hypertensive rats. Here, we determined the levels of O-methylated metabolites of the catecholamines, i.e., metanephrine, normetanephrine and 3-methoxytyramine, as proxy indicators for sympathetic activation. Plasma levels of O-methylated metabolites in SHR did not differ from those of WT (Fig. 3A). Likewise, the levels of Timp1—a circulating SASP that indicated ECM remodeling (Arpino, Brock & Gill, 2015). in SHR and WT rats were not different in all age groups (Fig. 3B). These data suggested that elevated blood pressure in our SHR model did not associate with widespread activation of sympathetic nervous system or systemic production of Timp1.

Figure 3 Plasma O-methylated metabolites of catecholamines and serum Timp1 in SHR.

(A) Plasma O-methylated metabolites of catecholamines, i.e., metanephrine (MET), normetanephrine (NMN) and 3-methoxytyramine (3MT), all of which are proxy indicators of widespread sympathetic activation, and (B) serum Timp1 in spontaneously hypertensive (SHR) and wild-type (WT) rats.

Upregulation of senescence-associated CDKI gene expression was found in heart and liver of SHR

The absence of systemic changes in Timp1 levels suggested that the onsets of hypertension-induced senescence were not synchronized among organs. Alternatively, susceptibilities to hypertension-induced senescence in those tissues were variable. Thus, we used the organ-based approach to dissect changes in the expression of various senescence-associated markers. We first focused on the heart since cardiac tissue remodeling and hypertrophy caused by pressure loads have been reported in hypertensive patients and animal models of essential hypertension (Lorell & Carabello, 2000; Pagan et al., 2022). Indeed, the expression of transcripts encoding two senescence-associated CDKIs, Cdkn2a and Cdkn1a, was strongly induced in the cardiac tissue of the 6-week-old SHR (Figs. 4A and 4E). This overt upregulation of Cdkn2a and Cdkn1a remained evident at 12, 24, and 36-week-old SHR (Figs. 4B–4D, 4F–4H). These increases in the expression of CDKI genes were consistent with the temporal profile of elevated blood pressure found in SHR.

Figure 4 Expression of inhibitors of cyclin-dependent kinases (CDKIs).

mRNA expression of inhibitors of cyclin-dependent kinases (CDKIs); Cdkn2a (A–D) and Cdkn1a (E–H) in the heart, liver, bone, kidney, and hypothalamus of SHR and WT rats at the age of 6, 12, 24 and 36 weeks. Results are presented as mean ± SEM. *P < 0.05; **P < 0.01; ***P < 0.001 compared with age-matched WT group. nd, not detected.

Identification of senescence markers in cardiac tissue but not in the blood of SHR implied that elevated blood pressure selectively induced premature aging process in certain tissues. To characterize these changes, assessment of Cdkn2a and Cdkn1a expression in the major vascular beds that were likely to be impacted by systemic hypertension, i.e., those of the liver, bone, kidney and hypothalamus. Indeed, we observed different changes in the pattern of Cdkn2a and Cdkn1a expression among these selected tissues (Figs. 4A–4H). Similar to the cardiac tissues, the expression of Cdkn2a and Cdkn1a mRNAs in the liver of SHR was significantly increased throughout the study. Consistently, an immunostaining analysis in hepatocytes showed an increase in p16 protein expression in SHR compared to WT (Fig. S2). On the other hand, these CDKI genes were downregulated in the hypothalamus of SHR. Likewise, the expression of Cdkn1a mRNA was significantly suppressed in the renal tissues. There was no change in Cdkn1a mRNA expression in bone tissues. Taken together, it was likely that chronic hypertension found in SHR initiated organ-specific senescence (i.e., heart and liver) via induction of CDKI expression.

Changes in SASP-related genes were consistent with the expression profile of CDKI gene expression in SHR

Next, we investigated the progression of senescence in these selected set of tissues by assessing the levels of SASPs. To further verify that induction of senescence in SHR was organ-specific, we evaluated the expression of Timp1 gene in those five organs. Indeed, mRNA levels of Timp1 gene were significantly upregulated in the heart and liver of SHR from the age of 6–36 months compared to age-matched WT rats (Figs. 5A–5D). Interestingly, the upregulation of Timp1 gene expression in SHR was markedly diminished by propranolol treatment (Fig. S3). Expression of Timp1 gene in bone revealed biphasic pattern, whereby the levels were initially decreased at 6 weeks and increased at 24 and 36 weeks of age. Furthermore, we found that the levels of Timp1 mRNA were significantly decreased in kidneys and hypothalamus in all agegroups. These data were similar to the pattern of CDKI mRNA expression. Figures 5E–5H show the levels of another matrix-associated senescence marker, matrix metallopeptidase 12 (Mmp12). The profile of Mmp12 gene expression in liver and kidneys (Figs. 5E–5H) were similar to those of Timp1 gene expression. Notably, the expression of Mmp12 gene was significantly increased in hepatic tissues of SHR. Levels of Mmp12 mRNA in renal tissue were decreased at 6 and 12 weeks old. These findings further support the association between elevated blood pressure and senescence phenotypes in heart and liver. In contrast, appearance of senescence phenotypes was delayed in tissues from kidneys and hypothalamus of SHR.

Figure 5 Expression of the senescence-associated secretory phenotypes (SASPs) markers.

mRNA expression of the senescence-associated secretory phenotypes (SASPs); Timp1 (A–D), and Mmp12 (E–H) in the heart, liver, bone, kidney, and hypothalamus of SHR and WT rats at the age of 6, 12, 24 and 36 weeks. Results are presented as mean ± SEM. *P < 0.05; **P < 0.01; ***P < 0.001 compared with age-matched WT group. nd, not detected.

Inflammatory cytokines, such as IL-6 and CXCL1, were also included as SASPs. We found an early increase in Il6 mRNA levels in the heart of 6–12-week-old SHR (Figs. 6A and 6B). Upregulation of Il6 gene was further identified in the liver of 24–36-week-old SHR (Figs. 6C and 6D). Expression of Il6 mRNA was significantly decreased in kidney of SHR (at 6, 12 and 36 weeks of age) and hypothalamus (at 36 weeks of age). Consistently, expression of Cxcl1 mRNA was markedly upregulated in the liver of SHR at 12, 24, and 36 weeks of age (Figs. 6E–6H). Moreover, we found that the levels of Cxcl1 transcripts were increased in bone of SHR at 36 weeks of age (Fig. 6H).

Figure 6 Expression of the senescence-associated secretory phenotypes (SASPs) markers.

mRNA expression of the senescence-associated secretory phenotypes (SASPs); Il6 (A–D), and Cxcl1 (E–H) in the heart, liver, bone, kidney, and hypothalamus of SHR and WT rats at the age of 6, 12, 24 and 36 weeks. Results are presented as mean ± SEM. *P < 0.05; **P < 0.01; ***P < 0.001 compared with age-matched WT group. nd, not detected.

SAXS/WAXS indicated misalignment of bone collagen fiber in SHR

Bone is a mineralized biological material that consists of cells embedded in the ECM. The mature mineral matrix is organized in a complex hierarchical network made from two major nanophases including collagen fibrils and hydroxyapatite, elongated platelet-like carbonated calcium phosphate particles (Hong et al., 2022). Because of the upregulation of Timp1 expression in the bone of SHR at the age of 24 and 36 weeks, it might also alter the ECM remodeling in the bone. Therefore, the collagen fiber alignment of the bone was determined by synchrotron SAXS/WAXS. As shown in Fig. 7A, SAXS analysis demonstrated a smaller peak at 1st through 6th orders in SHR, which suggested the poorly order and broad distribution of d-spacing in collagen fiber at 65.5 nm in SHR as compared to WT. In addition, the 2D-SAXS analysis showed the collagen oriented predominantly diagonal in WT, whereas SHR showed random orientation of collagen fiber (Fig. 7B). A ring that indicated collagen alignment in WT was more preferred orientation than that in SHR. Moreover, WAXS analysis of crystallite dimensions revealed a decrease in the crystal length and width in the bone of SHR (Figs. 7C–7D), which was calculated from FWHM of the (002) and (310) reflections (i.e., they indicated more crystallized collagen structure in WT as compared to SHR). Taken together, SHR was associated with the presence of disoriented collagen fibers and lower crystal dimensions in bone, which could be a consequence of hypertension and/or senescence.

Figure 7 SAXS/WAXS indicated misalignment of bone collagen fiber in SHR.

Simultaneous small- and wide-angle X-ray scattering (synchrotron SAXS/WAXS) analyses of collagen fibril arrangement and crystallite dimensions in the cortical bone specimens from 24-week-old WT and SHR. (A) 1st–6th peaks from SAXS analysis. (B) 2D-SAXS analysis. Arrowheads represent 1st and 3rd peaks corresponding to those indicated in Fig. 7A. (C) WAXS analysis of crystal width, and (D) WAXS analysis of crystal length. The measurement of the crystal width and length are presented in angstrom (Å) unit as mean ± SEM. *P = 0.067 compared with age-matched WT group (P < 0.1 is considered statistically significant due to the fact that a few angstrom changes in crystal length are considerable enough to affect bone matrix nanostructure).

Discussion

Hypertension has been described as a condition of premature vascular aging, especially endothelial cell senescence, relative to actual chronological age in both hypertensive patients and animal models of hypertension (McCarthy et al., 2019). There are various common pathways that play a role in cellular aging and blood pressure regulation, including but not limited to inflammation, oxidative stress, mitochondrial dysfunction, decreased klotho activity, activation of renin-angiotensin system as well as gut dysbiosis (Afsar & Afsar, 2023). Nonetheless, precise mechanisms by which hypertension accelerates premature aging and causes progressive end-organ damages remain largely elusive. Enhancing our understanding of cellular senescence could further be establish prognostic markers and diagnostic index of cardiovascular diseases, as well as provide alternative therapeutic targets for hypertension. The present study demonstrated a longitudinal study of mRNA expression of senescence biomarkers in the heart, liver, bone, hypothalamus, and kidney in SHR, an animal model of hypertension and relevant cardiovascular disease (Okamoto & Aoki, 1963; Reckelhoff et al., 1999). The heart is considered a blood pressure-loaded tissue. Previous investigations showed the classic paradigm of hypertensive heart disease with the left ventricular wall thickening in response to elevated blood pressure as a compensatory mechanism to minimize wall stress (Lorell & Carabello, 2000). An increase in oxidative stress is a major player in cardiac aging, and ROS production is a crucial process for accumulation of oxidative damages (Lorell & Carabello, 2000; Pagan et al., 2022). The elevation in the blood pressure induced circumferential wall stress and blood-flow-induced wall shear stress, which resulted in the degradation and reorganization of the ECM protein, such as collagen, elastin, proteoglycan and fibronectin (Lemarié, Tharaux & Lehoux, 2010). Aging of the heart is characterized by progressive impairment of heart function and the adaptive response that induces significant mechanical remodeling such as fibrosis, or accumulation of collagen and other ECM proteins (Izzo et al., 2021). Timp1, one of the markers of ECM remodeling in the cardiovascular system, is involved in the development of hypertension-mediated damage of the main target organs (Bisogni et al., 2020). An increase in Timp1 level was found to be associated with hypertension and left ventricular diastolic dysfunction (Hu et al., 2022). Our study showed an elevation of blood pressure in SHR from pre-adolescent period (6 weeks of age). However, the circulating level of Timp1 was unaltered in SHR despite cardiac mRNA expression of Timp1 being upregulated. It was suggested that, during aging, the senescence of stem and progenitor cells could hinder tissue remodeling by interfering with the ability of tissues to repair and regenerate.

According to the definition of senescence, there is the establishment of a stable growth arrest that limits the replication of damaged and senescent cells, in which cyclin and CDK complexes play an important role in controlling cell cycle (Beck, Horikawa & Harris, 2020; Chandler & Peters, 2013; Kumari & Jat, 2021). CDKIs including P16Ink4a and P21cip1, inhibit the kinase activities of the complexes and block transitions of the cell cycle at the early stage of senescence (Guo et al., 2022). The upregulation of P16Ink4a and P21cip1 in the heart and liver of SHR suggested a failure of these cells to continue dividing, which often occurs in cellular senescence. Consequently, senescent cells acquire an irreversible senescence-associated secretory phenotypes including secretion of soluble factors such as cytokines and chemokines, degradative enzymes like Mmps and Timps, and insoluble proteins/ECM components (Kumari & Jat, 2021). Elevation of Timp1 as well as Mmp12 in heart, liver, and bone could cause impaired regulation of connective tissue remodeling. Furthermore, increases in Il6 and Cxcl1 mRNA levels in the liver of SHR indicated the induction of a self- and cross-reinforced senescence and/or inflammation in the liver. Other cellular senescence markers such as CDKIs and SASPs were also upregulated in the liver of SHR. Liver is a key metabolic organ that regulates body energy metabolism and is metabolically connected to various tissues, including skeletal muscle, adipose tissue and pancreas (Rui, 2014). Aging has been shown not only to enhance vulnerability to acute liver injury but also increase the susceptibility of the fibrotic response (Kim, Kisseleva & Brenner, 2015). In addition, dysregulation of hepatic energy metabolism contributes to common age-related conditions, such as insulin resistance, diabetes mellitus, and non-alcoholic fatty liver disease (Marušić et al., 2021). It was suggested that cellular aging of hepatocytes in hypertension could promote tissue regeneration and function that accelerate impairment of energy metabolism.

Indeed, senescent cells and their biomarkers (e.g., p16) have been reported in various chronologically aging tissues from humans and animals (Table S1). Here, we also found higher expression of senescence markers prematurely in SHR, consistent with a report of senescence markers being detected and assessed in several tissue types (Tuttle et al., 2021). Both aging and hypertension also lead to vascular dysfunction and remodeling. Arterial changes in young hypertensive patients mimic those in aging normotensive individuals, indicating close interactions between biological aging and blood pressure elevation (Harvey, Montezano & Touyz, 2015; Nilsson, 2020; Wang & Bennett, 2012). In the present study, SHR exhibited thickening of aortic smooth muscle layer without noticeable change in endothelial cells. In addition, we found that an antihypertensive agent, i.e., propranolol, was able to diminish the upregulation of SASP, thus indicating an association between an increased cellular senescence and hypertension. Since premature aging was indeed observed and implicated in the development and progression of hypertension (Afsar & Afsar, 2023), it is possible that persistent elevation of blood pressure in essential hypertension is linked to the presence of premature vascular aging.

Senescent cells also secrete proteases, for example, matrix metalloproteinases (MMPs) that cause degradation ofECM. MMPs activity is regulated by endogenous protease inhibitors, the tissue inhibitors of metalloproteinases (TIMPs) (Arpino, Brock & Gill, 2015). It has been shown that the balance in the function of MMP and TIMP was disturbed in the process of premature aging (Cabral-Pacheco et al., 2020), which might also alter ECM remodeling. Besides their MMP inhibitory activity, TIMPs also participated in the regulation of various biological activities, such as cell growth, apoptosis, and differentiation independent of its metalloproteinase inhibitory activity (Lambert et al., 2004). After exposure to cytotoxic agents, initiation step of senescence in the cells is associated with expression of IL6 and TIMP1. Once senescence is established, SASPs are gradually released over a course of days up to months or years (Gilbert & Hemann, 2010; Sun, Coppé & Lam, 2018). Cardiac expression of TIMP1 and TIMP2 is significantly increased in chronic pressure-overloaded human hearts and is related to the degree of interstitial fibrosis (Heymans et al., 2005; Ishikawa, Hirose & Ishikawa, 2019). Apart from the cardiovascular system, hypertension is known to be associated with metabolic syndrome, which is characterized by insulin resistance, and strongly linked to the development of fatty liver (Nakagami, 2023; Rochlani et al., 2017). Furthermore, several clinical studies have demonstrated that hypertensive patients as well as animal models exhibited decrease in bone mineral density (BMD) with greater bone resorption than bone formation, thereby increasing age-related fracture risk (Li et al., 2016; Tiyasatkulkovit et al., 2019; Tsuda, Nishio & Masuyama, 2001). Bone is a highly dynamic and integrating tissue that is continuously resorbed by osteoclasts and formed by osteoblasts to maintain bone mass (Bolamperti, Villa & Rubinacci, 2022). Age-related bone loss involves a gradual and progressive decline in bone density. With increasing age, there is a significant reduction in bone formation and increased in bone resorption (Curtis et al., 2015). The contributing of cell senescence in mediating age-related bone loss has been demonstrated in both mice and humans. At least a subset of most cell types in the bone microenvironment becomes senescent with aging. Corresponding with the age-associated accumulation of senescent, osteocytes exhibited significantly higher expression of multiple SASP markers (Farr et al., 2016; Farr & Khosla, 2019). The increased TIMP levels contributed to the fibrotic buildup of ECM components (Arpino, Brock & Gill, 2015). In addition, TIMP1 functions as a suppressor of osteoblast growth and differentiation independent of its MMP inhibition capacity (Xi et al., 2020). Interestingly, elevated blood pressure was also reported to be associated with reduced bone mass and bone tissue structural deterioration, leading to bone fragility and higher risk of fractures (Hu et al., 2021; Tiyasatkulkovit et al., 2019). In the present study, increases in Timp1 and Cxcl1 mRNA levels together with the disoriented of collagen fibers and smaller crystal dimensions (Fig. 6) were found in bone tissues of SHR, suggesting that elevated blood pressure probably altered the ECM remodeling, leading to premature aging in bone tissue far beyond the chronological age (Turunen et al., 2016). On the other hand, upregulation of CDKIs and SASPs expressions were not observed in the hypothalamus and kidney, essential components in the regulation of arterial pressure. The molecular process of aging in the hypothalamus and kidney of SHR seems to be delayed. Although, the sensitivity of the hypothalamus to various feedback signals begins to decline with advanced age. The anti-aging effect modulated by exosomal miRNAs of hypothalamic neural stem cells (NSC) was reported (Zhang et al., 2017). In addition, it was reported that aging slowly altered the functions of hypothalamus and kidney through the regulation of the RE1-silencing transcription factor (REST), a repressor of neuronal genes during embryonic development. It is also a principal regulator of podocyte adaptation to injury and aging (Magassa et al., 2021). Podocytes are postmitotic cells that are exposed to different types of stress, including arterial pulse pressure. Increases in the expression and nuclear translocation of REST in podocytes could promote adaptation and resistance of cells to stress (Lu et al., 2014; Magassa et al., 2021). Our finding suggests that SHR were able to evade this hypertension-induced cellular senescence in the organs that can regulate blood pressure. However, these responses might be impaired in advancing age. The studied biomarkers are known to modulate immune and inflammatory responses. In addition, the compositions of CDKIs as well as SASP often vary depending on cell type and tissue of origin, and changes in their levels are able to represent the senescent processes (Lasry & Ben-Neriah, 2015; Maciel-Barón et al., 2016). Furthermore, the effects of sex and underlying genetic backgrounds should be determined in future in the context of senescence markers and tissue-specific premature aging. Future studies in female SHR as well as in other animal models of hypertension (e.g., high-salt diet treatment) should broaden our insights on how premature aging is linked to the development and progression of hypertension and related cardiovascular outcomes.

Conclusions

The present study investigated changes in the expression of senescence biomarkers in various organs in SHR. The cellular senescence profiles in our study suggested the presence of cellular senescence in some tissues (e.g., heart and liver), which could help explain the declines in cardiovascular function, impaired body metabolism, and a reduction in bone mass in SHR. Synchrotron SAXS and WAXS also revealed misalignment of collagen in the femoral cortical bone of SHR. Our findings, therefore, provide a foundation for developing strategies to modulate the senescence program for therapeutic benefits.

Supplemental Information

Supplemental Information 1 Reference list for the expression of senescence biomarkers in chronological aging in human and animal models.

The expression of senescence biomarkers in chronological aging in human and animal model in various organs and tissues.

Click here for additional data file.

Supplemental Information 2 Structural changes of aorta in SHR.

Representative H&E photomicrographs of aorta obtained from WT and SHR (A–B). Scale bars, 100 µ m. L, vascular lumen; TM, tunica media; TA, tunica adventitia. Representative TEM photomicrograph of cardiac tissue obtained from SHR at 3100x magnification. Scale bar, 2 µm. Ec, endothelial cell cytoplasm; M, mitochondria; Mu, muscle; RBC, red blood cell.

Click here for additional data file.

Supplemental Information 3 Expression of p16INK4a protein in liver cells of SHR.

Representative images of immunostaining against p16INK4a protein in liver cells of female SHR and WT. DAPI is a staining control and represents the nuclei of all cells. Red color visualized in the merged images represented colocalization of p16INK4a. Scale bars, 20 μm. Arrowheads indicate the nuclei that show co-localization of DAPI and p16INK4a signals.

Click here for additional data file.

Supplemental Information 4 Timp1 gene expression in SHR treated with propranolol.

The mRNA expression of Timp1 in the liver of SHR after being treated with a β-blocker agent (propranolol; Pro) for 12 weeks compared with the vehicle (Veh) group., *P< 0.05, compared with the vehicle control group.

Click here for additional data file.

Supplemental Information 5 Raw data of all experiments.

Click here for additional data file.

Supplemental Information 6 ARRIVE checklist.

Click here for additional data file.

The authors thank Professor Nateetip Krishnamra for critical comments on the manuscript, Dr. Sukanya Jaroenporn for technical advice about brain section, the scientific staffs at Faculty of Science, Burapha University for TEM, staffs at the Synchrotron Light Research Institute (SLRI), Nakhon Ratchasima, Thailand for technical assistance pertaining to SAXS/WAXS measurement at BL1.3W beamline, the National BioResource Project (NBRP), Institute of Laboratory Animals, Graduated School of Medicine, Kyoto University, Japan for the bio-resource through a memorandum of understanding with Mahidol University National Laboratory Animal Center (NLAC), Kanchana Kengkoom for coordination with NBRP, Arpakorn Pattanasawat, Pathnaree Wattano, and Jantanee Boonmameepool for technical assistance and Thitapha Kiattisirichai for artwork.

Additional Information and Declarations

Competing Interests

Author Contributions

Animal Ethics

Data Availability

The authors declare that they have no competing interests.

Ratthapon Somsura performed the experiments, analyzed the data, prepared figures and/or tables, authored or reviewed drafts of the article, and approved the final draft.

Kanokwan Kamkajon performed the experiments, analyzed the data, prepared figures and/or tables, authored or reviewed drafts of the article, and approved the final draft.

Khuanjit Chaimongkolnukul performed the experiments, prepared figures and/or tables, and approved the final draft.

Surachai Chantip performed the experiments, prepared figures and/or tables, and approved the final draft.

Jarinthorn Teerapornpuntakit analyzed the data, prepared figures and/or tables, authored or reviewed drafts of the article, and approved the final draft.

Kannikar Wongdee analyzed the data, prepared figures and/or tables, authored or reviewed drafts of the article, and approved the final draft.

Nuntaporn Kamonsutthipaijit performed the experiments, analyzed the data, prepared figures and/or tables, and approved the final draft.

Suwimol Tangtrongsup performed the experiments, analyzed the data, prepared figures and/or tables, and approved the final draft.

Nattapon Panupinthu conceived and designed the experiments, analyzed the data, prepared figures and/or tables, authored or reviewed drafts of the article, and approved the final draft.

Wacharaporn Tiyasatkulkovit conceived and designed the experiments, performed the experiments, analyzed the data, prepared figures and/or tables, authored or reviewed drafts of the article, and approved the final draft.

Narattaphol Charoenphandhu conceived and designed the experiments, analyzed the data, authored or reviewed drafts of the article, and approved the final draft.

The following information was supplied relating to ethical approvals (i.e., approving body and any reference numbers):

Institutional Animal Care and Use Committee (IACUC), Faculty of Science, Mahidol University.

The following information was supplied regarding data availability:

The raw data of all experiments is available in the Supplemental Files.

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
