# Peer review of "Tissue-specific expression of senescence biomarkers in spontaneously hypertensive rats: evidence of premature aging in hypertension"

_PeerJ, doi:10.7717/peerj.16300_

## Round 0.1 · original submission · Major Revisions

You will find enclosed three sets of reviewers comments. These indicate that extra work is required and some modifications/amendments to the paper are needed. These are extensive and all must be carefully and fully addressed. Without this extra data requested by the reviewers (and supported by me) the manuscript will not be accepted.

As noted by both reviewers 1 and 3, the effect of an anti-hypertensive agent should be examined to provide support for your conclusions. This is essential.
Are there sex differences and if so, how are they manifest?
A more direct link between senescence and ageing is required for the conclusions to be valid.
Are there changes in other tissues (reviewer 3).
Changes in mRNA should be validated by some attempt to quantify protein expression levels using immunoblot or ELISA approaches.

Reviewer 1 ·

Basic reporting

This manuscript clearly compared the differences in cellular senescence biomarkers, including CDKIs and SASPs, between SHR and normotensive Wistar rats. It also provided the explicit results of their mRNA and protein levels. The current findings are not enough to support its conclusion to deem that persistently elevated blood pressure causes cellular senescence in some organs. At least, it is a little hard to conclude the causative relationship between hypertension and cellular senescence.

Experimental design

Some comments:
1. The supporting experiments are necessary to prove that anti-hypertensive therapeutic strategies (some treatments) may reverse the elevated Timp1 level in SHR.
2. Fig.4 shows that Timp1 is no difference between 6 weeks and 36 weeks among WT. it indicates that 36 weeks rats are not aging rats. In the instruction section. It is mentioned that hypertension is a major health problem in the aging population. If the aging group will be added, it will highlight the meaningfulness of anti-hypertension in aging patients.
3. It is undoubtedly known that the reason for SHR should be some gene to be missense. The young rats also shew high blood pressure. The high-salt diet or other hypertension models also induce similar results.
4. Age- is mentioned in the title. The corresponding results were not posted to compare among ages. Or aging wants to be emphasized in the concept.

Validity of the findings

no comment

Additional comments

the findings in this manuscript are very meaningful to guide the cellular senescence research in hypertension and aging. it needs more evidence to support the current conclusion. or to support the causative relationship between hypertension and cellular senescence.

Reviewer 2 ·

Basic reporting

no comment

Experimental design

no comment

Validity of the findings

no comment

Additional comments

The editorial office of Peer j

Thanks the editorial office of Peer j invited me to review the manuscript of [peerj-86625], I had reviewed manuscript,my opinion list blow.

It is interesting that Cellular senescence was identified in an organ, especially, the organs directly affected by high blood pressure or functional defecting organs. The present study provides experimental evidence that hypertension affects people’s life span which is relate to Cellular senescence existing in important organs or end-organs. The manuscript that logical arrangement is clear. But the minor errors need to be corrected.
Minor errors
In the page 3, line 81. whether the clause“ DNA damage reactive oxygen species (ROS),” need to be add a comma between DNA damage and reactive oxygen species (ROS),
In the page 5, line 176. Is the “ΔCt method” the same “ 2-ΔΔCt method”?. Please explain it.
In the page 9, line 335-337, The sentences “The heart is considered a blood pressure-loaded tissue.Previous investigations showed that hypertensive heart exhibited ventricular, accumulation of oxidative stress, excess production of ROS, and the aging-related stressors” is  so hard to understand, suggest that make it sample sentence. Heart is an organ but a tissue.

Thanks
Best Regard
Sincerely

Reviewer 3 ·

Basic reporting

See below.

Experimental design

See below.

Validity of the findings

See below.

Additional comments

General comments:

This manuscript describes experiments that investigated changes in the expression of senescence biomarkers in various organs of SHR. The results suggested the presence of cellular senescence in some tissues (e.g., heart and liver), which could explain the declines in cardiovascular function, impaired body metabolism, and a reduction in bone mass in SHR. SAXS and WAXS analysis revealed misalignment of collagen in the femoral cortical bone of SHR.

Although the study had some interesting results, there are several things that need to be answered. The following should be considered:

1. Are these differences between males and females?
2. What happens in the vasculature (arteries and veins) – smooth muscle and endotheilium?
3. What happens if the SHR are placed on an antihypertensive agent?
4. Are there changes in the adrenal glands? Sympathetic nerves?

The following reference should be noted:

A Ansari et al. Sex- and age-related differences in renal and cardiac injury and senescence in stroke-prone spontaneously hypertensive rats. Biol Sex Differences 14, article number 33, 2023.

Specific comments:

Abstract and Introduction: Please state a hypothesis.

Line 244: “evaluate” should be “evaluated”.

Line 262: “was” should be “were”.

Line 295: “showed” should be “show”.

Line 300: “supported” should be “support”.

Line 315: Delete “an”.

Line 360: “cytokine” should be “cytokines”.

Line 382: “altered” should be “alter”.

Line 419: “advance” should be “advanced”.

Line 427: Delete “in”.

Line 428: “regulation” should be “regulate” and delete “of”.

Line 440: “provided foundation” should be “provide a”.

---

## Round 0.2 · accepted · Accept

Thank you for carefully addressing and/or discussing the issues raised. I am pleased to recommend acceptance of your paper.

Reviewer 1 ·

Basic reporting

The authors have answered all my comments. So far, I do not have new comments about this manuscript.

Experimental design

The authors have answered all my comments. So far, I do not have new comments about this manuscript.

Validity of the findings

The authors have answered all my comments. So far, I do not have new comments about this manuscript.

Additional comments

The authors have answered all my comments. So far, I do not have new comments about this manuscript.

Reviewer 3 ·

Basic reporting

Acceptable.

Experimental design

Acceptable.

Validity of the findings

Acceptable.

Additional comments

Acceptable.